# IDV Typer: An Automated Tool for Lineage Typing of Influenza D Viruses Based on Return Time Distribution

**DOI:** 10.3390/v16030373

**Published:** 2024-02-28

**Authors:** Sanket Limaye, Anant Shelke, Mohan M. Kale, Urmila Kulkarni-Kale, Suresh V. Kuchipudi

**Affiliations:** 1Bioinformatics Centre, Savitribai Phule Pune University (Formerly University of Pune), Pune 411007, India; sanket.limaye22@gmail.com (S.L.); anantshelkecell@gmail.com (A.S.); 2Department of Statistics, Savitribai Phule Pune University (Formerly University of Pune), Pune 411007, India; mohan.kale80@gmail.com; 3Department of Infectious Diseases and Microbiology, University of Pittsburgh School of Public Health, Pittsburgh, PA 15261, USA

**Keywords:** Influenza D virus, lineage typing, IDV Typer server, return time distribution

## Abstract

Influenza D virus (IDV) is the most recent addition to the *Orthomyxoviridae* family and cattle serve as the primary reservoir. IDV has been implicated in Bovine Respiratory Disease Complex (BRDC), and there is serological evidence of human infection of IDV. Evolutionary changes in the IDV genome have resulted in the expansion of genetic diversity and the emergence of multiple lineages that might expand the host tropism and potentially increase the pathogenicity to animals and humans. Therefore, there is an urgent need for automated, accurate and rapid typing tools for IDV lineage typing. Currently, IDV lineage typing is carried out using BLAST-based searches and alignment-based molecular phylogeny of the hemagglutinin-esterase fusion (HEF) gene sequences, and lineage is assigned to query sequences based on sequence similarity (BLAST search) and proximity to the reference lineages in the tree topology, respectively. To minimize human intervention and lineage typing time, we developed IDV Typer server, implementing alignment-free method based on return time distribution (RTD) of k-mers. Lineages are assigned using HEF gene sequences. The server performs with 100% sensitivity and specificity. The IDV Typer server is the first application of an RTD-based alignment-free method for typing animal viruses.

## 1. Introduction

Influenza D virus (IDV) is the newest member of the *Orthomyxoviridae* family, classified under the *Deltainfluenzavirus* genus [1]. The virus, like other Orthomyxoviruses, is a negative stranded, RNA virus with a segmented genome. IDV was first isolated in 2011 from pigs in Oklahoma [2]. Since then, IDV has been isolated/detected from cattle across many countries, including the United States (US), Mexico [3], Ireland [4], United Kingdom [5], France [6], Italy [7], China [8], Canada [9], Japan [10], Turkey [11], Netherlands [12], Brazil [13], South Korea [14], Australia [15], Argentina [16], Namibia [17] and Denmark [18]. Serological evidence also indicates IDV presence in Morocco, Togo, Benin, Kenya [19], Luxembourg [20] and Ethiopia [21]. While cattle are considered the main reservoir, serological evidence of IDV infection was found in several animals, including pigs, sheep, goats, camels, giraffe, wildebeest, white-tailed deer and horses [3,6,7,8,17,19,22,23,24,25,26,27]. IDV’s pervasive presence among US cattle is strikingly evident. IDV antibodies also have been detected in US cattle sera from 2003 to 2004 [28]. Further, a study conducted between 2014 and 2015 found IDV seropositive cattle samples in 41 out of 42 states nationwide. An overall seropositivity rate of 77.5% was recorded among tested cattle sera samples. Notably, this prevalence exhibited considerable regional variation, ranging from 47.7% to 84.6% in different parts of the country. These findings underscore the widespread prevalence of IDV within US cattle populations [29].

Many viral respiratory pathogens have been implicated in Bovine Respiratory Disease (BRD) including Bovine Viral Diarrhea Virus (BVDV), Bovine Herpesvirus type 1 (BHV-1), Bovine Respiratory Syncytial Virus (BRSV), Bovine Parainfluenza Virus 3 (BPIV-3) and Bovine Coronavirus (BCoV) [30]. Other viruses such as bovine adenovirus (BAdV), bovine rhinitis A virus and bovine adeno-associated virus have been found to be associated with BRD less frequently [31,32]. In addition to these existing pathogens, a growing body of evidence suggests a correlation between IDV in the pathogenesis of BRD [3,33,34,35]. Notably, D/CA2019 lineage, the most recently identified IDV lineage in the US, was isolated from calves with respiratory disease [36]. The presence of serological evidence [37,38] and the detection of IDV nucleic acid in human nasal samples [39,40] collectively constitute compelling proof of IDV infection in humans, underscoring its zoonotic potential.

The IDV genome is segmented, and the seven segments consist of three polymerases, i.e., Polymerase basic protein 2 (PB2), Polymerase basic protein 1 (PB1) and Polymerase protein (PA/P3); Hemagglutinin-Esterase Fusion (HEF) protein, Nucleoprotein (N), Matrix protein (M); and the Non-structural protein (NS). The HEF protein is primarily involved in receptor binding and is the target for neutralizing antibodies [2]. HEF is a type 1 transmembrane protein containing one long ectodomain, a transmembrane region, one small cleavable N-terminal signal peptide and a short cytoplasmic tail. Sialic acid(SA) serves as the main receptor for IDV HEF. IDV utilizes either 9-*O*-acetylated Neu5Ac (Neu5,9Ac_2_) and Neu5Gc (Neu5Gc9Ac), independent of the α2,3 or α2,6 linkage, as a receptor for entering the cell [41].

IDV is currently classified into five major lineages, namely, D/OK, D/660, D/Yama2016, D/Yama2019 and D/CA2019 based on the nucleotide sequence divergence of the HEF gene [36,42,43,44]. The D/OK and D/660 lineage isolated from the USA and Italy are reported to undergo reassortment events with each other [42,45]. The two lineages, D/Yama2016 and D/Yama2019 were reported from Japan [43,44] and recently the D/Yama2019 lineage was reported from China, South Korea and Australia [14,15,46]. Even though IDV was only discovered a decade ago, novel genetic clusters and entire lineages are being discovered [44] including multiple and frequent reassortments [42,47]. In Hokkaido, Japan, the viruses isolated from two out of three BRD outbreaks in cattle during 2018–2020 belonged to a new lineage, different from the lineages of Japanese IDVs based on HEF, PB2, P3, NP and NS genes [48]. The viruses in the new lineage were isolated from cattle with severe disease including rapid spread in the herd, debilitation and death due to pneumonia. This study found coinfection of IDV with other bacterial and viral pathogens, highlighting the role of IDV as one of the causative pathogens of BRD. Further, the IDVs of the new lineage had poor cross-neutralization with sera against heterogenous IDV, indicating the circulation of genetically and antigenically distinct IDV lineages in Japan, D/Yama2016 and D/Yama2019. Some IDV isolates from Japan are genetically distinct from IDV from the rest of the world [10].

A new US antigenic lineage was discovered when the first IDV sequences from California were sequenced [36]. These IDVs form a new clade named D/CA2019 in which the P3 gene groups in the D/OK lineage and the other genes cluster with the D/660 lineage. Moreover, sera raised to D/OK or D/660 had poor neutralizing titers against D/CA2019 [36]. The D/OK and D/660 IDV reassortments of P3 and NP gene were also reported from the province of Quebec, Canada [9]. An IDV isolate similar to swine and bovine IDV in the US was sequenced in France in 2012 [6] and reported to form a separate cluster from the clusters of known lineages. This isolate was annotated as lineage D/Fra [49] or D/France-2012 [50] by different groups.

Accurate classification of the circulating lineages could help in tracking the evolution of IDV and the potential impact on animal and human health. In addition, rapid and precise typing methods help in disease diagnosis and surveillance efforts and facilitate basic and translational research. Owing to the diversification of the IDV into multiple lineages, typing IDV is currently carried out using alignment-based phylogeny methods [36,42,43,44]. The lineages are assigned based on clustering proximity of the unknown with the known ones, and most often than not, the typing analyses need to be carried out in an iterative mode, which requires considerable amount of time. Similarly, lineage assignment to the query sequence is also carried out using BLAST-based searches wherein the lineage of the sequence sharing highest identity and similarity with end-to-end query coverage is assigned to the query. However, the lack of lineage information in the GenBank entries poses a challenge for assignment of lineage to the query sequence using BLAST-based methods. Evolutionary changes in the IDV genome have resulted in the expansion of genetic diversity and the emergence of multiple lineages that might expand host tropism and potentially increase the pathogenicity to animals and humans. Therefore, there is an urgent need for automated, accurate and rapid typing tools for IDV lineage typing.

This paper reports the development of the IDV Typer webserver using an alignment-free method developed to automate the process of typing for IDV [51,52]. The alignment-free method is based on the return time distribution (RTD) concept and corresponding statistics that enables clustering, molecular phylogeny and typing. This RTD method has been previously implemented for the development of typing servers for viruses such as Mumps virus (MuV), Dengue virus (DENV), West Nile virus (WNV) and Human Rhino virus (HRV) [53,54,55,56], which provides a distinctive advantage for typing of the said viruses over alignment-based methods. Typing servers for human enteroviruses, human rhinoviruses, noroviruses, Zika, dengue, chikungunya and yellow fever viruses and SARS-CoV-2 employing alignment-based methods such as BLAST-based searches have also been developed and are available online [57,58,59,60].

## 2. Materials and Methods

### 2.1. Datasets

The IDV HEF gene sequences obtained from GenBank and the literature review were curated and compiled as three independent datasets such as reference, true positive and true negative, which were used to develop an alignment-free RTD-based method for lineage typing of IDV.

### 2.2. Reference Dataset

A total of 23 HEF gene sequences of IDV obtained from GenBank with sequence length in the range of 1796–1995 bp (≥90% query coverage) were compiled, for which the lineage information of the sequences was obtained from the literature [2,15,18,36,45,46,49]. The list of accession numbers, corresponding lineage information and reference (PubMed ID) is provided in Appendix A.

### 2.3. True Positive Dataset

The HEF gene sequences were extracted from GenBank by performing a BLAST search using Refseq entry (Accession number: NC_036618.1) as the query. The sequences used in the reference dataset were excluded from the list of significant hits, and thus a total of 143 entries were obtained. The lineage information for 142/143 entries was compiled from the literature review [11,12,15,17,18,36,45,46,50,61].

The lineage information was assigned to the remaining entry using alignment-based Maximum-Likelihood (ML) phylogeny. The 143 entries (142 entries with known lineage and one entry with unknown lineage) of the True Positive dataset were aligned using the MAFFT algorithm implemented in SEED 2 software [62]. Based on the Bayesian Information Criterion (BIC) implemented in the IQ-Tree webserver [63,64], the TVM+F+G4 nucleotide substitution model was used for generating an ML-based phylogenetic tree, and lineage was assigned to the entry (with unknown lineage) based on clustering proximity with other isolates (with known lineage information). The accession numbers, lineage information and reference (PubMed ID) of the 143 entries used as the True Positive dataset are listed in Appendix A. This dataset was used to test the sensitivity of the typing server.

### 2.4. True Negative Dataset

The negative dataset consisting of 50 entries was generated for testing the specificity and sensitivity of the typing server. The HEF and non-HEF gene sequences from other *Orthomyxoviridae* family members with lengths in the range of 1796–1995 bp were used in the dataset. The non-HEF gene sequences of the IDV were also included in this dataset. The GenBank accession numbers, gene names and virus names of 50 entries used in the True Negative Dataset are listed in Appendix A.

### 2.5. Method

The steps involved in the development of the RTD-based typing server are: (a) reference sequence dataset curation and compilation, (b) optimization of k-mer, (c) calculate the RTDs and their parameters for the optimum value of k, (d) obtain a pairwise Euclidean distance matrix based on the parameters of RTD obtained in (c), (e) generation of a Neighbor-joining (NJ) tree using the Euclidean distance matrix, (f) derivation of inter-lineage distance cut-off ranges, (g) assign each lineage to the sequences based on nearest reference lineage and respective distance cut-off range of a specific lineage. Steps (a) and (b) are essential for the development of virus-specific typing servers.

### 2.6. k Value Optimization for IDV Typing

Optimization of *k* is an essential step in the RTD-based typing method to achieve accuracy, sensitivity and specificity [52]. The protocol for optimization of *k* for IDV data is as follows. The statistical parameters, mean (µ) and standard deviation (σ) of each RTDs (4^k^) for every *k*-mer in the range of 1 to 8 were calculated for every sequence in the reference dataset [53] and hence, every gene sequence in the reference dataset constituted a numeric vector of 2 × 4^k^ dimensions. Numeric vectors for each value of k were used for the generation of a Euclidean-based pairwise distance matrix that was used to infer Neighbor-joining (NJ)-based phylogeny trees [52,65] by using the DendroPy library [66] in Python. The optimum *k* value was chosen based on the accuracy of clustering in accordance with the lineage classification of the reference dataset.

### 2.7. IDV Typing Server Development

The lineage typing server for IDV, named “IDV Typer” was developed using Apache, CGI and PHP architecture. The server is available in the public domain at http://bioinfo.unipune.ac.in/IDV/home.html

## 3. Results

The IDV reference dataset was compiled to include representative sequences of the five known lineages. The lineages D/Yama2016, D/Yama2019 and D/CA2019 included five, five and three members, respectively, hence two sequences each from D/Yama2016, D/Yama2019 and D/CA2019 were included in the reference dataset. The D/France2012 lineage (tentative sixth lineage) included in reference dataset consisted of a single entry D/bovine/France/2986/2012 (Accession ID: LN559126) as it forms an independent branch outside the other known lineages [49,50]. Thus, a reference dataset of 23 entries was used as a training dataset for optimization of k and designing of RTD-based lineage typing method for IDV.

In the case of the reference dataset of IDV, RTD-based NJ trees were derived for different values of *k* (1 to 8). At *k* = 5 (Figure 1), lineage-wise clustering was observed and no misclassification was detected, indicative of the optimum *k* value (*k* = 5).

Though complete HEF gene sequence is recommended for lineage typing, analysis of performance of IDV Typer indicated that a minimum coverage of ≥90% (≥1796 bp) of sequence length of HEF gene is required.

### 3.1. “IDV Typer” RTD Method-Based Server for Lineage Typing of IDV

“IDV typer” (http://bioinfo.unipune.ac.in/IDV/home.html), an RTD-based lineage typing server for IDV was developed and uses the reference dataset along with the derived parameters as a backend knowledgebase. The server takes HEF gene sequences with >90% of sequence coverage in FASTA format through a sequence submission form. The CGI scripts at the back end convert sequences into RTD vectors and compute the statistical parameters (µ and σ) for the 1024 pentamer RTDs for the query sequence/s and calculate the Euclidean distance between every query sequence and the sequences of the reference dataset. The lineage assignment to the query sequence is carried out using a 2-step procedure. The first step involved the identification of the lineage of the closest reference sequence based on a distance measure, and the second step involved verifying if the distance of the query sequence from the sequence of respective reference lineage, lay within the pre-computed distance ranges. Successful lineage identification is subject to meeting both these criteria. Failure to meet these criteria for all the lineages results in a message being returned as “No lineage can be assigned to this query sequence using RTD-based method” on the IDV typer page. The time required for the lineage assignment for a dataset consisting of 15–100 sequences is 1–5 s.

### 3.2. Server Validation

The server’s performance was assessed using the True Positive and True Negative datasets compiled as reported in the Materials section. The True Positive dataset consists of 143 entries of IDV HEF gene sequences. The lineage information was assigned to these sequences either based on the literature survey (#142) or clustering proximity (#1) in alignment-based phylogeny with the True Positive dataset. The information regarding number of entries from every lineage used in the Reference and True Positive datasets is provided in Table 1. The one entry for which lineage information was not available was assigned lineage D/660 based on the clustering proximity in the ML-based tree. One entry isolated from Turkey (GenBank ID: OP353622) was reported to form an independent branch and was tentatively assigned a novel lineage called D/Bursa2013 [11]. However, it was observed that this entry formed an independent branch within the D/OK lineage in the ML-based phylogenetic tree of the True Positive dataset. Therefore, this entry was assigned as D/OK lineage. Phylogenetic analysis by a recent study reported independent branching of few isolates and classified them as independent groups. These groups were named as D/Michigan-2019, consisting of two entries (GenBank ID: MW632174 and MW079478) and D/Texas-2017, consisting of one entry (GenBank ID: MT636473) [50]. However, these isolates have been classified under the D/OK lineage by other recent studies [15,46]. Therefore, these isolates have been assigned to D/OK lineage in this study. The lineage assignment of all 143 entries by the RTD-based IDV typer server corroborated with the lineage assignment carried out by literature review and alignment-based ML methods, which confirmed 100% sensitivity of the IDV Typer server.

The negative dataset consisted of 50 entries from the members of the *Orthomyxoviridae* family. The RTD-based alignment-free method performed with 100% specificity as it did not assign any lineage to the entries of the True Negative dataset.

## 4. Discussion

Genomic surveillance is a cornerstone in proactively monitoring and managing emerging viral pathogens. In particular, proactive surveillance of emerging pathogens like IDV, which exhibits a propensity for genetic diversity and cross-species transmission, is crucial. Accurate typing of such viral variants facilitates the elucidation of their evolutionary trajectories and the identification of potential virulence factors and determinants of host specificity. Moreover, it enables real-time tracking of viral spread and informs the development of targeted interventions, ranging from vaccine design to surveillance strategies. Therefore, genomic surveillance coupled with precise typing methodologies is indispensable in safeguarding public health and mitigating the impacts of emerging viral threats. Since the isolation of IDV from swine in 2011 in Oklahoma, USA, its widespread prevalence across multiple countries and its ability to infect numerous animal hosts have become evident [22,23,24]. The spread of the virus on such a large scale, along with developments in sequencing technologies, has led to increased sequencing of virus samples, facilitating the understanding of IDV evolution and genetic diversity.

The threat of IDV to public health looms large due to its ability to infect not only humans but also multiple non-animal hosts. With a global distribution and the capacity to evolve continually, IDV presents a formidable challenge, particularly exacerbated by the lack of regular testing and surveillance efforts. Without ongoing monitoring, the virus can mutate and spread under the radar, potentially leading to the emergence of lineages with increased pathogenic and/or virulence in humans, leading to epidemics and or pandemics. The global circulation of IDV and its ability to infect a wide range of hosts has prompted the need to detect and map diversifying lineages, which requires developing a system for identifying and classifying the virus into their respective lineages. Automated, accurate and rapid IDV typing tools can help in the quick identification of currently circulating and emerging lineages.

The phylogenetic and evolutionary studies of IDV have been carried out to understand diversification of IDV into known lineages. These studies provide insight into the evolution of IDV and the emergence of new lineages. A study has reported the presence of a lineage-specific HEF amino acid mutation that might impact antibody binding affinity [42]. Consequently, a recent study has established the broader antigenicity of the D/CA2019 lineage over the others as antibodies elicited by D/CA2019 isolates might neutralize isolates of other lineages [36]. The presence of sites under positive selection with lineage specific mutations in HEF has also been reported [18,67]. With the ability to type circulating IDV lineages rapidly, the IDV Typer server can help in studies characterizing the potential impact on host tropism, human health implications and pathogenicity.

The most frequently used approach for IDV lineage typing is based on alignment-based phylogenetic analysis of HEF gene sequences, which involves multiple sequence alignment (MSA). Even though the currently available alignment-based molecular phylogeny analyses (MPA) method is useful, it has two major limitations, the complexity of analysis increases significantly with an increase in the size of the dataset and the need for human intervention for contextual interpretation of tree topologies to assign lineage. Furthermore, with the addition of even a single entry, the whole analysis, beginning with MSA needs to be repeated. The BLAST-based approach is also occasionally used for assignment of lineage to IDV isolates. However, BLAST-based lineage typing is subject to both manual intervention and availability of annotation/metadata in the knowledgebase. Currently, GenBank entries of significant number of isolates lack lineage information and therefore, there is a need to perform downstream analysis involving alignment-based phylogeny or extensive literature survey in order to assign lineages to the query sequences from respective isolates.

To overcome these limitations, a novel alignment-free RTD-based method for fast and accurate lineage typing of IDV is developed. Customization of RTD approach for typing of any virus requires the development of a virus-specific knowledgebase (reference dataset) and the derivation of optimum value of *k* [52] that helps to determine the pre-computed empirical cut-off and eliminates the basic step of MSA implied in the alignment-based method for geno/sero/lineage typing. Using 1024 pentamers of RTDs for alignment-free MPA of reference and true positive sequences was completed within a few minutes and needed to be completed only by developers. A comparative analysis using a True Positive dataset was carried out to identify the time required for lineage typing of IDV sequences using RTD-based IDV Typer, BLAST-based search and alignment-based MPA. The IDV typer was able to assign correct lineage information to 143 entries from True Positive dataset in about 12 s. In comparison, the BLAST-based search (online mode) and alignment-based phylogeny (standalone mode) of the True Positive dataset required an execution time of about 42 and 500 s, respectively, which is further extended due to the need for manual interpretation of results for lineage assignment. Thus, the RTD-based IDV Typer required significantly less amount of time for correct lineage assignment to 143 entries which indicates the high speed and accuracy of RTD-based typing as compared to other methods. The knowledgebase is updated periodically by the developers and therefore, in case of identification of a novel lineage and/or reclassification isolates to different lineage, the whole process of updating the knowledgebase, k-mer optimization and deriving genetic distance cut-off will be carried out by the developers and users need not worry about it and can carry out lineage assignment without having to compile reference datasets and perform alignment-based phylogenetic analysis.

Thus, the development of virus-specific RTD-based alignment-free typing servers addresses three gaps, namely, the elimination of the step involving MSA for MPA, automation of typing protocol and a significant reduction in computational time required for typing while maintaining 100% accuracy. This proves to be a unique strength of the RTD-based alignment-free method over the conventional alignment-based method; hence, for faster surveillance of circulating lineages of IDV, the RTD-based server was developed.

## 5. Conclusions

The alignment-free RTD-based method with suitable modifications was successfully applied for the lineage typing of IDV. A web server implementing the alignment-free RTD-based method was developed and made available online for the assignment of the IDV into the known lineages without human intervention and thereby automating the lineage assignment process. The IDV Typer server is a highly sensitive, specific and computationally efficient tool that provides an alternative over conventional methods such as alignment-based MPA and BLAST-based searches. The IDV Typer outperforms available methods for lineage assignment in terms of computational efficiency and essential accuracy. It automates the process of lineage tying while providing an easy to use interface, due to which the tool could be deployed in the field for real-time surveillance of the circulating IDV lineages and help to monitor genetic diversity of IDVs isolates. The IDV typer can, therefore, be a handy tool for better understanding the epidemiology of IDV and enabling genomic surveillance.

## Figures and Tables

**Figure 1 viruses-16-00373-f001:**
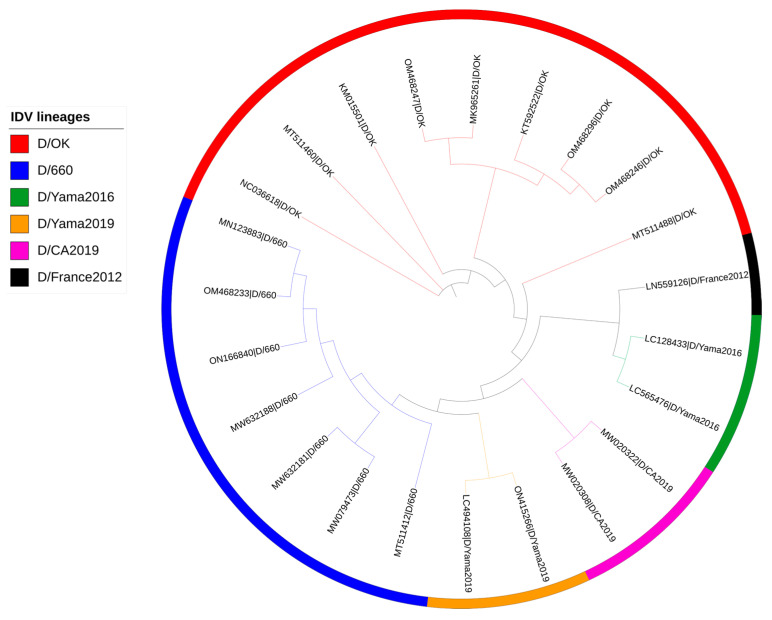
Alignment-free RTD-based phylogenetic tree of HEF gene reference dataset for *k* = 5. The tip labels are divided into two parts based on the pipe (|) character. The first part indicates GenBank accession numbers and the second part indicates lineage. The tree was visualized and annotated using the ITOL server (https://itol.embl.de/).

**Table 1 viruses-16-00373-t001:** Lineage-wise distribution of reference dataset.

Lineage	Reference Dataset	True Positive Dataset
D/OK	9	89
D/660	7	47
D/Yama2016	2	3
D/Yama2019	2	3
D/CA2019	2	1
D/France2012	1	0

## Data Availability

The original contributions presented in the study are included in the article and Appendix A.

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
