# Peer review of "IDV Typer: An Automated Tool for Lineage Typing of Influenza D Viruses Based on Return Time Distribution"

_viruses, 2024, doi:10.3390/v16030373_

Round 1
Reviewer 1 Report (Previous Reviewer 1)
Comments and Suggestions for Authors
The present manuscript from Limaye et al. describes a new automated alignment-free tool that was developed in order to facilitate the typing of influenza D virus lineages. The authors improved the quality of the manuscript compared to the previous version, I have however the following minor comments:
Line 15: the first sentence should be rephrased as “Influenza D virus (IDV) is the most recent addition to the Orthomyxoviridae family and cattle serve as the primary reservoir.”
Line 19: “Lieages” should be replaced by “lineages”.
Line 33: The authors should review the text throughout the whole manuscript to correct the inconsistent use of the space before citations.
Lines 35-37: The introduction should be more precise about IDV distribution. The authors should include the complete and updated list of countries where IDV was detected, South America is missing from the list (Brazil, Argentina), and also Africa (Morocco, Togo, Ethiopia, Namibia, Kenya) as well as Australia, South Korea, Netherlands, Luxembourg, Turkey. Please, either provide the complete list of the countries, or state that IDV was detected on all continents.
Line 43: “(BHV-)1” should be replaced with “(BHV-1)”.
Line 43: The authors should include at least one reference describing the respiratory pathogens involved in BRD that were included in this sentence.
Lines 60-62: IDV belonging to the Yamagata clades is actually an emerging genotype in Asia, it has been indeed already detected in China, Japan, and recently in South Korea (Lim et al., 2023) and also Australia (Brito et al., 2023). Since the manuscript is focused on IDV lineages the authors should clarify this in the introduction.
Line 62: Please delete one of the two dots at the end of the sentence.
Line 104: Please delete one of the two dots at the end of the sentence.
Line 119: once again, more than 180 IDV HEF complete codifying sequences have been released on NCBI. Why did the authors download only a part of them to validate the IDV typer tool?
Line 240: The authors describe here the advantages of using the RTD-based method for fast typing, but the discussion should include a few sentences also about the disadvantages of using this method compared to Maximum-Likelihood model and alignment-based methods in general.
Line 262: I agree with the authors that this method is indeed fats and very useful for fast IDV typing, especially since the number of new IDV sequences released is accumulating fast in public databases. I however think that the discussion should also include the future potential limitations of this method. Will the authors update the true positive for each IDV genotype? What would happen if in two years some sequences (among the ones used to validate the method) were to be re-classified within a different genotype? The authors need to address this issue for better clarity to the reader.
Comments on the Quality of English LanguageOnly minor editing of English is required
Author Response
Reviewer 1:
Line 15: the first sentence should be rephrased as “Influenza D virus (IDV) is the most recent addition to the Orthomyxoviridae family and cattle serve as the primary reservoir.”
Thank you for the suggestion. We have made the necessary change to the manuscript
Line 19: “Lieages” should be replaced by “lineages”.
Thank you for the suggestion. We have made the necessary changes to the manuscript.
Line 33: The authors should review the text throughout the whole manuscript to correct the inconsistent use of the space before citations.
Thank you for the suggestion. We have made the necessary changes to the manuscript.
Lines 35-37: The introduction should be more precise about IDV distribution. The authors should include the complete and updated list of countries where IDV was detected, South America is missing from the list (Brazil, Argentina), and also Africa (Morocco, Togo, Ethiopia, Namibia, Kenya) as well as Australia, South Korea, Netherlands, Luxembourg, Turkey. Please, either provide the complete list of the countries, or state that IDV was detected on all continents.
Thank you for the suggestion. We have made the necessary changes to the manuscript and added the citations.
Line 43: “(BHV-)1” should be replaced with “(BHV-1)”.
Thank you for the suggestion. We have made the necessary changes to the manuscript.
Line 43: The authors should include at least one reference describing the respiratory pathogens involved in BRD that were included in this sentence.
Thank you for the suggestion. We have added the citation in the manuscript.
Lines 60-62: IDV belonging to the Yamagata clades is actually an emerging genotype in Asia, it has been indeed already detected in China, Japan, and recently in South Korea (Lim et al., 2023) and also Australia (Brito et al., 2023). Since the manuscript is focused on IDV lineages the authors should clarify this in the introduction.
Thank you for the suggestion. We have made the necessary changes to the manuscript and added the citations.
Line 62: Please delete one of the two dots at the end of the sentence.
Thank you for the suggestion. We have made the necessary changes to the manuscript.
Line 104: Please delete one of the two dots at the end of the sentence.
Thank you for the suggestion. We have made the necessary changes to the manuscript.
Line 119: once again, more than 180 IDV HEF complete codifying sequences have been released on NCBI. Why did the authors download only a part of them to validate the IDV typer tool?
Thank you for the question, however, as of 29-01-2024, there are a total of 189 IDV HEF sequences with length ≥90% (≥1796 bp) of total length (1995bp) present in GenBank. These entries were curated to eliminate sequences having ambiguous bases (characters other than ATGC) as RTD does not such sequences. A total of 166 entries were retained of which 23 entries were used as reference dataset and 143 entries as True Positive dataset.
Line 240: The authors describe here the advantages of using the RTD-based method for fast typing, but the discussion should include a few sentences also about the disadvantages of using this method compared to Maximum-Likelihood model and alignment-based methods in general.
RTD has several advantages as compared to BLAST-based and alignment-based Maximum-Likelihood method which have been mentioned in the manuscript (line 274 to 279 and 287 to 292). The only apparent disadvantage of RTD is non-inclusion of bootstrap values for tree topology. The premise of RTD is based on the order and frequency in which the characters and hence k-mers appear in the sequences. RTD uses summary statistics of mean and standard deviation to cluster the sequences. Therefore, conventional bootstrap method which involves shuffling of columns in MSA cannot be used in RTD. We have validated application of block bootstrap (reshuffling blocks of sequences rather than shuffling columns) to validate essential correctness and accuracy of branching topologies in the trees generated using RTD (Kolekar, et al., 2012). However, we would also like to note that the typing is done on the basis of statistical confidence of extent of variations within and between lineages which is mentioned in manuscript (line 212 to 218).
Line 262: I agree with the authors that this method is indeed fats and very useful for fast IDV typing, especially since the number of new IDV sequences released is accumulating fast in public databases. I however think that the discussion should also include the future potential limitations of this method. Will the authors update the true positive for each IDV genotype? What would happen if in two years some sequences (among the ones used to validate the method) were to be re-classified within a different genotype? The authors need to address this issue for better clarity to the reader.
Thank you for the suggestion. The authors will be updating the True Positive dataset periodically and if information about emergence of new lineage and/or assignment of isolates to different lineage is found in the literature, the same will be updated in the knowledgebase. This has been mentioned in the manuscript at line 293 to 298.
Note: In response to comments by reviewer 3, we revisited all the datasets and made the following changes to reference, True Positive and True Negative datasets:
- One entry from True Positive dataset (GenBank ID: ON166840) belonging to D/660 lineage was shifted to Reference dataset.
- Seven entries from True Negative dataset (GenBank IDs: MW096808, DQ997471, KJ848679, MK969543, MN507189, D63468 and MK965333) were observed to contain ambiguous bases and hence were replaced (GenBank IDs of new entries: MW220337, OR762342, MZ717369, CY182406, OQ997395, KM504281 and FR671423)

Reviewer 2 Report (Previous Reviewer 2)
Comments and Suggestions for Authors
Authors are quite eloquent in responding to my comments.
Authors insist that their RTD method is perfect for influenza D.
Author Response
Authors are quite eloquent in responding to my comments.
Authors insist that their RTD method is perfect for influenza D.
Thank you.
Reviewer 3 Report (New Reviewer)
Comments and Suggestions for Authors
Review of IDV Typer: An automated tool for lineage typing of Influenza D viruses based on return time distribution
Limaye et al. are providing a paper describing a webserver tool able to determine the lineage of influenza D viruses HEF segment sequence using a fast, RTD-based, alignment-free method.
As a general comment of the paper, while the computational aspect of the described RTD-based method seem to be proven and while the webserver works, with few issues however (see below), the method itself is not described enough for readers that do not know how RTD-based approaches are working. Moreover, the tool has several flaws that need to be addressed before the paper can be accepted for publication.
1- The authors claim (L235 et L236) that "With the ability to detect novel IDV lineages rapidly, the IDV Typer server can help in studies characterizing the potential impact on host tropism, human health implications, and pathogenicity." and moreover L261 "for faster surveillance of emerging and circulating lineages of IDV".
However, the IDV Typer webserver has not been tested with HEF sequences that are different enough from the 5 major clades the tool tests the sequences for. The authors thus cannot claim that their tool is able to identify "novel IDV lineages" or "emerging IDV lineages". To prove it, the authors should have reference sequences of one of the lineages left out from the reference dataset. These sequences should then be tested by the tool to see if the IDV Typer attributes these highly divergent sequences to one of the lineages ; or rather to the "D/unassigned" category, as it should, meaning that the tested sequence is from a new IDV lineage that is not present in the reference set.
To stress this important issue a bit more, the authors claim that IDV are currently classified into 5 major lineages: D/OK, D/660, D/Yama2016, D/Yama2019 and D/CA2019 according to the references 23, 28, 29 and 30. However, the authors are not citing another important paper describing few more IDV lineages, namely D/Shandong-2014, D/Texas-2017, D/Michigan-2019 and D/Quebec-2020, according to Gaudino et al. 2022: "Evolutionary and temporal dynamics of emerging influenza D virus in Europe (2009–22)" published in Virus Evolution. The authors must most likely review their method at the light of this paper and support these clades on top of the already supported ones. This paper from 2022 should be cited in introduction and most likely discussed. To support the importance of this comment and the previous one, if we look at the D/bovine/Quebec/3M-B/2020 HEF sequence (MT246289). According to Gaudino et al. 2022 this sequence corresponds to a specific small genogroup between the D/660 and D/OK lineages. If we look at the provided Table S2, this sequence (noted MT246289) is shown two times and classified both once as D/660 and once as D/OK in the Table S2. Firstly, it is extremely strange to have the same sequence tested two times in the Table S2. Secondly, that same sequence got two different lineages attributed: D/660 and D/OK. Considering this sequence is from a specific clade between D/660 and D/OK, this clearly shows that IDV Typer is not able to properly attribute to the “D/unassigned” lineage the sequences that are coming from a clade different enough from the tool reference database. This makes sense since the "D/unassigned" is simply based on one sequence, D/bovine/France/2986/2012. Such an issue is extremely dangerous for the IDV surveillance as it will “hide” emergent lineages under the pre-determined lineages of IDV Typer instead of the "D/unassigned" lineage. The tool in its current state would thus most likely damage the IDV surveillance community as a whole, and this issue must thus be addressed before acceptance for publication.
2- Another major issue that must be addressed is the report of statistics of confidence for the lineage attribution to a given sequence. If a sequence is submitted and is different enough to be determined as a new clade based on phylogenetic methods, IDV Typer will still loosely attribute a given clade to the sequence without giving the user any information with which level of confidence it was attributed to the said lineage, which refers to the previous issue.
3- The authors also do not provide enough proofs for the attribution of the sequences to the correct lineage. They should thus provide a table of sequence reference tested, the classification given by IDV Typer, and the classification given by the litterature with the article reference. The Table S2 corresponds to this, but no litterature reference is given to know if the IDV Typer attributed genogroup is correct or not. Such a table should also be available on the IDV Typer website.
4- Computation statistics such as wall-clock time or CPU time using the same computation resources (allocated CPU threads and RAM) of RTD-based methods compared to BLAST methods or phylogeny methods (MAFFT+IQTREE2) are not reported. This could nicely support the authors’ claims about RTD-based methods speed for IDV HEF lineage classification, especially since the authors ran phylogenetic analyses with MAFFT and IQ-Tree to assign IDV genogroups to sequences of the True Positive Dataset.
5- The authors do not provide enough information to justify that k=5 was the most optimal parameter for their RTD-based method, or why the RTD method they developed requires a minimum of coverage of 90% of the HEF gene.
6- In the discussion, the authors are only comparing RTD methods to the computationally intensive phylogenetic methods to determine lineages of viral segments. However, BLAST-based genotyping methods are often used to assign genogroups to sequences and also do not need multisequence alignments to be performed (not in the sense of phylogenetic MSAA). The authors should thus compare - or at the very least discuss - their method versus regular BLAST-based genotyping methods to support the speed and accuracy of their RTD method.
7- In the discussion, the authors are not pointing one of the major weakness of the RTD-based methods, being their abscence of flexibility. If a new clade is detected and needs to be added to the reference dataset, the whole RTDs, distance matrix and NJ phylogenies computations, as well was the k value optimisation, need to be calculated and tested again. Meanwhile BLAST-based genotyping methods are highly flexible since new sequences can be added easily to the reference set of sequences, and easily provide results with understandable confidence scores (percentage of identity, sequence alignment coverage, bitscore, etc), something IDV Typer does not provide at the moment.
8- While I used the webserver, I faced an issue that many users will most likely have and this needs to be fixed before acceptance of the publication. The server only accepts sequences in capital letters, which can be issue depending on the sequences processing tools and pipelines used by the webserver users. A support for lowercase sequences by the IDV Typer webserver is necessary so that it can be used comfortably. Moreover, gap characters (-) are also triggering errors. These should be handled by the IDV Typer webserver as this can be an issue for users managing their sequences with MEGA, which is a very broadly used fasta files and alignments manager.
9- In the introduction, the authors report the importance of reassortants on internal genes concerning IDV evolution, and yet the webserver presented in the paper only classifies HEF sequences. This should be highlighted as a limit of the tool in the discussion or the conclusion.
10- The abstract does not provide sufficiently clear information concerning why alignment-based molecular phylogeny or sequence identification by BLAST are a "criticial gap", since they are the current gold standards. Moreover, the abstract does not provide enough information about the "k=5" and why it is relevant.
11- The naming convention of the IDV strains in the main text are not respected, they should be under the form of D/host/country/identifier/year.
12- As a minor comment, Line 62 and 104 there are two consecutive dots (..).
Author Response
Reviewer 3:
As a general comment of the paper, while the computational aspect of the described RTD-based method seem to be proven and while the webserver works, with few issues however (see below), the method itself is not described enough for readers that do not know how RTD-based approaches are working. Moreover, the tool has several flaws that need to be addressed before the paper can be accepted for publication.
1- The authors claim (L235 et L236) that "With the ability to detect novel IDV lineages rapidly, the IDV Typer server can help in studies characterizing the potential impact on host tropism, human health implications, and pathogenicity." and moreover L261 "for faster surveillance of emerging and circulating lineages of IDV".
Thank you for this suggestion. We have removed the words “novel” and “emerging” from manuscript.
However, the IDV Typer webserver has not been tested with HEF sequences that are different enough from the 5 major clades the tool tests the sequences for. The authors thus cannot claim that their tool is able to identify "novel IDV lineages" or "emerging IDV lineages". To prove it, the authors should have reference sequences of one of the lineages left out from the reference dataset. These sequences should then be tested by the tool to see if the IDV Typer attributes these highly divergent sequences to one of the lineages ; or rather to the "D/unassigned" category, as it should, meaning that the tested sequence is from a new IDV lineage that is not present in the reference set.
Thank you for this suggestion. Since, the isolate from France has not yet been assigned any lineage, we had annotated as D/unassigned, which appears to have indicated very different message that the server can identify and annotate isolates of novel lineage as “D/unassigned”. In response to this, two changes have been made: (a) Isolate from France (GenBank: LN559126) is reannotated as D/France2012 (line 177) on the basis of publications (Gaudino, et al., 2022; Wan, et al., 2020). (b) D/unassigned category has been removed. With these changes the server is expected to assign either one of the known lineages to a query sequence (HEF) or state that “No lineage can be assigned”, thereby ensuring lineage status clarity to the user.
To stress this important issue a bit more, the authors claim that IDV are currently classified into 5 major lineages: D/OK, D/660, D/Yama2016, D/Yama2019 and D/CA2019 according to the references 23, 28, 29 and 30. However, the authors are not citing another important paper describing few more IDV lineages, namely D/Shandong-2014, D/Texas-2017, D/Michigan-2019 and D/Quebec-2020, according to Gaudino et al. 2022: "Evolutionary and temporal dynamics of emerging influenza D virus in Europe (2009–22)" published in Virus Evolution. The authors must most likely review their method at the light of this paper and support these clades on top of the already supported ones. This paper from 2022 should be cited in introduction and most likely discussed. To support the importance of this comment and the previous one, if we look at the D/bovine/Quebec/3M-B/2020 HEF sequence (MT246289). According to Gaudino et al. 2022 this sequence corresponds to a specific small genogroup between the D/660 and D/OK lineages. If we look at the provided Table S2, this sequence (noted MT246289) is shown two times and classified both once as D/660 and once as D/OK in the Table S2. Firstly, it is extremely strange to have the same sequence tested two times in the Table S2. Secondly, that same sequence got two different lineages attributed: D/660 and D/OK. Considering this sequence is from a specific clade between D/660 and D/OK, this clearly shows that IDV Typer is not able to properly attribute to the “D/unassigned” lineage the sequences that are coming from a clade different enough from the tool reference database. This makes sense since the "D/unassigned" is simply based on one sequence, D/bovine/France/2986/2012. Such an issue is extremely dangerous for the IDV surveillance as it will “hide” emergent lineages under the pre-determined lineages of IDV Typer instead of the "D/unassigned" lineage. The tool in its current state would thus most likely damage the IDV surveillance community as a whole, and this issue must thus be addressed before acceptance for publication.
Thank you for this suggestion, however, based on the Maximum-Likelihood phylogenetic tree (Figure 2 in Gaudino, et al., 2022), the D/Shandong-2014 and D/Quebec-2020 isolates have been classified as independent groups under the D/OK and D/660 lineages (Gaudino, et al., 2022). The D/Texas-2017, D/Michigan-2019 have been classified by other groups (Brito, et al., 2023; Yu, et al., 2022) as independent branches designated by respective authors as groups within the D/OK lineage. D/bovine/Quebec/3M-B/2020 (D/Quebec-2020; GenBank MT246289) sequence is present only once in the True positive dataset and assigned lineage D/660.
2- Another major issue that must be addressed is the report of statistics of confidence for the lineage attribution to a given sequence. If a sequence is submitted and is different enough to be determined as a new clade based on phylogenetic methods, IDV Typer will still loosely attribute a given clade to the sequence without giving the user any information with which level of confidence it was attributed to the said lineage, which refers to the previous issue.
The premise of RTD is based on the order and frequency in which the characters and hence k-mers appear in the sequences. RTD uses summary statistics of mean and std. to cluster the sequences. Therefore, conventional bootstrap method which involves shuffling of columns in MSA cannot be used in RTD. We have validated application of block bootstrap (reshuffling blocks of sequences rather than shuffling columns) to validate essential correctness and accuracy of branching topologies in the trees generated using RTD (Kolekar, et al., 2012). However, we would also like to note that the typing is done on the basis of statistical confidence of extent of variations within and between lineages which is mentioned in manuscript (line 212 to 218).
3- The authors also do not provide enough proofs for the attribution of the sequences to the correct lineage. They should thus provide a table of sequence reference tested, the classification given by IDV Typer, and the classification given by the litterature with the article reference. The Table S2 corresponds to this, but no litterature reference is given to know if the IDV Typer attributed genogroup is correct or not. Such a table should also be available on the IDV Typer website.
Thank you for the suggestion. We have added an extra column to the Supplementary tables 1 and 2 which includes the PubMed ID (PMID) of literature used for assigning lineage to a particular sequence. These two updated tables have been made available on IDV Typer website.
In view of your suggestion, we revisited all the datasets and have made the following changes:
- One entry from True Positive dataset (GenBank ID: ON166840) belonging to D/660 lineage was shifted to Reference dataset.
- Seven entries from True Negative dataset (GenBank IDs: MW096808, DQ997471, KJ848679, MK969543, MN507189, D63468 and MK965333) were observed to contain ambiguous bases and hence were replaced (GenBank IDs of new entries: MW220337, OR762342, MZ717369, CY182406, OQ997395, KM504281 and FR671423)
4- Computation statistics such as wall-clock time or CPU time using the same computation resources (allocated CPU threads and RAM) of RTD-based methods compared to BLAST methods or phylogeny methods (MAFFT+IQTREE2) are not reported. This could nicely support the authors’ claims about RTD-based methods speed for IDV HEF lineage classification, especially since the authors ran phylogenetic analyses with MAFFT and IQ-Tree to assign IDV genogroups to sequences of the True Positive Dataset.
Thank you for this suggestion. We have added the appropriate statements in discussion section (line 274 to 279 and 287 to 292).
5- The authors do not provide enough information to justify that k=5 was the most optimal parameter for their RTD-based method, or why the RTD method they developed requires a minimum of coverage of 90% of the HEF gene.
Thank you for this suggestion. At k=5, all the sequences in the reference dataset formed clusters according to their lineages and there was no misclassification. Hence, k=5 was chosen as optimal k. Text in manuscript has been suitably modified in result section (line 184 to 187).
The users are recommended to use complete HEF sequence (1995 bp) for lineage typing. However, it is often observed that sequencing platforms may have ambiguous/missing bases. Therefore, we have validated minimum length requirement for correct typing and found that minimum 90% of sequence coverage is required to maintain accuracy of lineage typing. The section 2.7 of Method ("Sequence length optimization”) has been removed to avoid confusion. However, the statements (line 199 to 202) in the result section are retained with suitable modifications.
6- In the discussion, the authors are only comparing RTD methods to the computationally intensive phylogenetic methods to determine lineages of viral segments. However, BLAST-based genotyping methods are often used to assign genogroups to sequences and also do not need multisequence alignments to be performed (not in the sense of phylogenetic MSAA). The authors should thus compare - or at the very least discuss - their method versus regular BLAST-based genotyping methods to support the speed and accuracy of their RTD method.
Thank you for this suggestion. We have added the appropriate statements in discussion section (line 274 to 279 and 287 to 292).
7- In the discussion, the authors are not pointing one of the major weakness of the RTD-based methods, being their abscence of flexibility. If a new clade is detected and needs to be added to the reference dataset, the whole RTDs, distance matrix and NJ phylogenies computations, as well was the k value optimisation, need to be calculated and tested again. Meanwhile BLAST-based genotyping methods are highly flexible since new sequences can be added easily to the reference set of sequences, and easily provide results with understandable confidence scores (percentage of identity, sequence alignment coverage, bitscore, etc), something IDV Typer does not provide at the moment.
Thank you for this suggestion. We partially agree with you that BLAST-based lineage typing is open ended subject to availability of new lineage data in the knowledgebase. However, the assignment of lineage is subject to manual intervention and availability of annotation/metadata in the knowledgebase. Based on our experience we would like to put forward two limitations regarding knowledgebase (lack of lineage annotation to isolates) and need to perform alignment-based phylogenetic analysis for lineage assignment to isolates. The authors will be updating the True Positive dataset periodically and if information about emergence of new lineage and/or assignment of isolates to different lineage is found in the literature, the same will be updated in the knowledgebase. This has been mentioned in the manuscript at line 293 to 298.
8- While I used the webserver, I faced an issue that many users will most likely have and this needs to be fixed before acceptance of the publication. The server only accepts sequences in capital letters, which can be issue depending on the sequences processing tools and pipelines used by the webserver users. A support for lowercase sequences by the IDV Typer webserver is necessary so that it can be used comfortably. Moreover, gap characters (-) are also triggering errors. These should be handled by the IDV Typer webserver as this can be an issue for users managing their sequences with MEGA, which is a very broadly used fasta files and alignments manager.
Thank you for the suggestions. The issue of server not accepting sequences in lowercase has been resolved.
Gap (-) characters are only added to sequences post alignment. As this is an alignment-free method, users are expected to upload sequence entries without carrying out alignment. Similarly, the presence of ambiguous base will make the base uninformative and therefore, sequences with ambiguous bases cannot be used for lineage typing. Therefore, instructions have been mentioned on the sequence submission page that the query sequences should not contain any ambiguous bases and gap characters.
9- In the introduction, the authors report the importance of reassortants on internal genes concerning IDV evolution, and yet the webserver presented in the paper only classifies HEF sequences. This should be highlighted as a limit of the tool in the discussion or the conclusion.
Lineage is assigned to an IDV isolate only on the basis on HEF gene sequence and therefore the server uses only HEF gene sequence for lineage assignment. HEF qualifies to be marker for lineage assignment since no reassortment are reported in HEF. Though, reassortments are observed in internal genes, it does not impact lineage assignment to IDV.
10- The abstract does not provide sufficiently clear information concerning why alignment-based molecular phylogeny or sequence identification by BLAST are a "criticial gap", since they are the current gold standards. Moreover, the abstract does not provide enough information about the "k=5" and why it is relevant.
Thank you for the suggestion. The abstract text is suitably modified.
11- The naming convention of the IDV strains in the main text are not respected, they should be under the form of D/host/country/identifier/year.
Thank you for the suggestion. We have updated the naming conventions in the manuscript
12- As a minor comment, Line 62 and 104 there are two consecutive dots (..)
Thank you for the suggestion. We have made the necessary changes in the manuscript.

Round 2
Reviewer 2 Report (Previous Reviewer 2)
Comments and Suggestions for Authors
I can only support that this manuscript being published as a "short communication".
My judgement have also been echoed by other reviewers.
Author Response
We made significant improvements to the introduction and discussion sections. as recommended by the reviewer.
We deeply value and respect the opinion of the reviewer. As mentioned earlier, due to the constraints of a short communication format, we are unable to include sufficient methodological details. Therefore, we kindly request that this manuscript be considered for publication as a regular research article, as we believe it holds significant value for the scientific community. Thank you for your understanding and consideration.
Reviewer 3 Report (New Reviewer)
Comments and Suggestions for Authors
The authors appropriately answered my interrogations and brought the necessary changes to the tool. There is still few minor points to address:
1- The authors are claiming that the sequence LN559126 has been reannotated as D/France2012, but when I tested this sequence on the IDV Typer Website on the 19/02/2024, it returned D/unassigned as a result, and not D/France2012. It is not a big problem, as long as it didn't return erroneous lineage like D/660 or D/OK for that sequence.
I was not able to test sequences that would give the "No lineage can be assigned" result.
I agree with the authors about the D/Shangdong2014 and D/Quebec2020 lienages being assigned to D/OK and D/660 respectively.
2- I agree with the authors that RTD-based methods are adequate. However, some statistics of confidence of attribution of a given sequence to a given lineage should be provided on the website results page/output so that the user can assess if the uploaded sequence could be part of a potentially new clade close to an existing clade.
3- I thank the authors for the changes that should improve the submitted manuscript.
4- I thank the authors for the changes that should improve the submitted manuscript.
5- The following section mentionned by the authors "2.7. Sequence Length Optimization" in the Methods is missing from the re-submitted manuscript and should thus be added.
6- I want to thank the authors for the added information in the manuscript about BLAST-based methods compared to RTD-based methods.
7- I partially agree with the authors.
8- Some FASTA management softwares can add gaps at the end of sequences after saving the FASTA file (MEGA, AliView). This can be issue for certain users if the server does not automatically remove gap characters (-) from the submitted sequences. The updated instructions for sequence upload on the webserver should be enough however.
9- I agree with the authors.
10, 11 and 12- I want to thank the authors for the improved abstract.
Line 25: two consecutive commas.
Line 283: is poorly written.
Please revise the english of the added paragraphs, for instance (but not limited to) L274 to 279 and L287 to 292.
Author Response
The authors appropriately answered my interrogations and brought the necessary changes to the tool. There is still few minor points to address:
1- The authors are claiming that the sequence LN559126 has been reannotated as D/France2012, but when I tested this sequence on the IDV Typer Website on the 19/02/2024, it returned D/unassigned as a result, and not D/France2012. It is not a big problem, as long as it didn't return erroneous lineage like D/660 or D/OK for that sequence.
Thank you for the comment. Due to some technical glitch the server had crashed because of which it reverted to the last available version. We however have fixed this issue now.
I was not able to test sequences that would give the "No lineage can be assigned" result.
You may please test the server with any of the entry in the true negative dataset and or by downloading HEF of Influenza C (which is closest, yet a different virus).
I agree with the authors about the D/Shangdong2014 and D/Quebec2020 lienages being assigned to D/OK and D/660 respectively.
Thank you.
2- I agree with the authors that RTD-based methods are adequate. However, some statistics of confidence of attribution of a given sequence to a given lineage should be provided on the website results page/output so that the user can assess if the uploaded sequence could be part of a potentially new clade close to an existing clade.
Thank you. As mentioned in the previous reply, the server cannot designate a sequence to a new lineage as it uses a knowledgebase of existing/known lineages for the purpose of complete automation of lineage typing process. The authors, need to update the reference knowledge set as and when new lineage/s are designated for IDV.
3- I thank the authors for the changes that should improve the submitted manuscript.
Thank you for the comment.
4- I thank the authors for the changes that should improve the submitted manuscript.
Thank you for the comment.
5- The following section mentionned by the authors "2.7. Sequence Length Optimization" in the Methods is missing from the re-submitted manuscript and should thus be added.
Sequence optimisation was an internal exercise to determine lowest length of sequence that can be correctly types using the server, which turned out to be 90%. Therefore, the section was removed but the 90% cutoff criteria for sequence length is retained in the legend to Figure 1.
6- I want to thank the authors for the added information in the manuscript about BLAST-based methods compared to RTD-based methods.
Thank you.
7- I partially agree with the authors.
Thank you for understanding.
8- Some FASTA management softwares can add gaps at the end of sequences after saving the FASTA file (MEGA, AliView). This can be issue for certain users if the server does not automatically remove gap characters (-) from the submitted sequences. The updated instructions for sequence upload on the webserver should be enough however.
Thank you for the comment.
9- I agree with the authors.
Thank you for the comment.
10, 11 and 12- I want to thank the authors for the improved abstract.
Thank you for the comment.
Comments on the Quality of English Language
Line 25: two consecutive commas.
Thank you for the comment. We have updated it
Line 283: is poorly written.
Thank you for the comment. We have revised it
Please revise the english of the added paragraphs, for instance (but not limited to) L274 to 279 and L287 to 292.
Thank you for the comment. We have revised these two paragraphs.

This manuscript is a resubmission of an earlier submission. The following is a list of the peer review reports and author responses from that submission.
Round 1
Reviewer 1 Report
Comments and Suggestions for Authors
The present manuscript from Limaye et al. describes a new automated alignment-free tool that was developed in order to facilitate the typing of influenza D virus lineages. The tool could be very useful as currently no official IDV nomenclature is available, unlike for other influenza viruses. In addition, it is indeed true that assigning a sequence to a clade could be computationally heavy as in the future the number of IDV sequences will likely increase and alignment-based methods require a certain amount of time. I just have some major concerns about the quality of the typing method described in the present paper, which would require in my opinion further optimization. Please see the following comment that will hopefully be considered to improve the quality of the manuscript:
1) The method is based on the computation of a distance matrix, which is widely outperformed by likelihood methods. While distance methods are indeed faster, they do not take into account the evolutionary history of a sequence, making them much less powerful. The authors however fail to include this important element in the discussion section. IDV typer is a very interesting tool, it should however be “advertised” as a fast-typing tool and a clear statement about the limits of this methods should be present in the manuscript.
2) Another major flow is the complete absence of a precise method/criteria to define a “genotype”. Unfortunately, a literature search is not a good method as different labs could have different methods to define an IDV genotype. One manuscript in literature attempted to define some criteria for IDV genotyping (Gaudino et al., 2022). Before developing “IDV typer tool” the authors should therefore define themselves IDV genotypes by setting accurate criteria for IDV typing based on robust methods such as likelihood or Bayesian inference. For instance, some sequences such as LN559126, MW632174.1 or MT636473.1 have been defined as genetically divergent from clade D/OK (Wan et al., 2020, Ducatez et al., 2015). Therefore, based on which criteria did the authors define these sequences as belonging to D/OK clade?
3) In the introduction, the authors state several times that reassortment events are widely present in influenza D viruses. Why did the authors decide to base the typing only on HEF then?
4) To date, more than 180 IDV HEF complete codifying sequences have been released on NCBI. Why did the authors download only a part of them to validate the IDV typer tool?
Minor comments:
Line 35: in the majority of the cited studies IDV was only molecularly detected and not isolated. Please substitute the word “isolated” with “detected” or with “detected or isolated”.
Line 41: to date IDV is not considered a significant contributor to BRD, as discussed in several articles present in literature. IDV is detected in clinically ill animals but often in coinfection with other respiratory pathogens. Experimentally, IDV was shown to have a clinical impact only with Mycoplasma bovis coinfection (Lion et al., 2021) but not with Mannheimia haemolytica (Zhang et al., 2019). As IDV impact in BRD is still under discussion, this sentence should be rephrased.
Line 45-46: IDV nucleic acids were indeed detected in human samples (Borkenhagen et al., 2018; Leibler et al., 2023), although in clinically healthy people. Please rephrase this sentence.
Lines 47-51: these sentences should be substantiated by at least one reference.
Line 55: in reference number 7, no reassortment events between IDV lineages were described. Please revise the references.
Lines 52-54: currently, no official classification is present for IDV, unlike for other influenza viruses. The lineages described in literature (D/OK, D/660… etc) are based on the topological clustering of different clades upon phylogenetic analyses. The authors should rephrase this sentence.
Line 59: the reference 28 does not describe the discovery of a novel IDV lineage, as the authors could retrieve only 664 bp of the HEF sequence. This reference is therefore misleading, please revise it.
Lines 62-64: by reading this sentence one might think that IDV caused pneumonia and death among the herd. The animals were however coinfected with other pathogens (BCoV, BPIV-3) including bacteria (M. bovis, M. haemolytica, P. multocida), which are the main responsible for pneumonia pathogenicity. Please include these elements in the sentence.
Lines 69-70: this sentence is redundant with the sentence in lines 57-58. Please revise it.
Lines 73-74: the same level of details should be given to all the reassortant IDV described. Please add which segments were reassorted also for IDV detected in Canada.
Comments on the Quality of English Language
Minor English editing required
Reviewer 2 Report
Comments and Suggestions for Authors
Authors tried to apply an alignment-free methods, based on RTD, to "lineage typing" the IDV found particularly in bovine.
1. as listed in lines 90-96, this RTD-based analysis is not novel as it is claimed, since it had been applied in the study of wide variety of animal viruses.
2. the results are too preliminary. If the tool is as powerful as classical alignment-based phylogeny methods, authors should at least compare this IDV typer with more regular alignment-based methods, regarding several validating criteria, such as sensitivity, specificity, and others. In other words, you need a golden standard or positive control.
3. as shown in Figure 1, so far IDV has only 5 lineages and in this manuscript a limited dataset of 21 sequences were analyzed based on RTD, it is too preliminary to claim that this methods works well for IDV. To test this methods, I encourage authors to work more challenging data set, like avian influenza or human influenza virus which have an huge dataset and more numerous clades or lineages or genomic subtypes.
4. line 173, to require a minimum coverage of >90% (>1796 bp) in order to run this RTD-methods is a severe limitation. In nature or viruses, There are numerous genes that are much smaller than 1.8 kb, and regular practice in diagnosis is usually based on partial sequences of said a few hundred bp, so this methods won't be welcome by lab diagnosticians.
5. even if it is powerful or rapid regarding statistics, wet lab nucleotide sequence work are still required and is a much necessary prerequisite. This is simply off-bench desk work.
6. other classical alignment-based methods are also automated. They also have a number of soft wares to support.
7. Even if you have typed the IDV lineage in seconds, There is nothing you can do clinically regarding treatments or prevention or for vaccination.